# Adhesives free bark panels: An alternative application for a waste material

**Charlett Wenig**[1], **Friedrich Reppe**[1], **Nils Horbelt**[1], **Jaromir Spener**[2], **Ferréol Berendt**[2], **Tobias Cremer**[2], **Marion Frey**[3], **Ingo Burgert**[3], **Michaela Eder**[1] *

1 Department of Biomaterials, Max Planck Institute of Colloids and Interfaces, Potsdam, Germany,
2 Department of Forest Utilization and Timber Markets, Faculty of Forest and Environment, Eberswalde University for Sustainable Development, Eberswalde, Germany, 3 Wood Materials Science, Institute for Building Materials, ETH Zürich, Zürich, Switzerland

* Michaela.Eder@mpikg.mpg.de

**Data Availability Statement:** The data is now accessible here: https://edmond.mpdl.mpg.de/dataset.xhtml?persistentId=doi:10.17617/3. AZRWF3.

## Abstract

The proportion of bark in tree trunks is in the range of ~ 10–20%. This large amount of material is currently mainly considered as a by- or even waste-product by the timber processing industry. Recently, efforts towards the use of bark have been made, e.g. as a raw material to harvest different chemical compounds or as an additive for wood particle boards. Our motivation for this work was to keep the bark in an almost natural state and explore alternative processes and applications for use. The traditional method of de-barking tree trunks by peeling was used to harvest large bark pieces. Two pieces of peeled bark were placed crosswise, with the rhytidom side (outer bark) facing each other. After different conditioning steps, bark pieces were hot pressed to panels without adding adhesives. These experiments on bark samples of different Central European tree species suggest that production of panels with species dependent properties is possible and feasible. This is a step towards producing sustainable panels by using a natural waste material, while retaining its beneficial structure and its natural chemical composition.

## Introduction

In ancient times, the material utilization of bark as a by-product of wood was a common way to make most efficient use of the whole tree trunk. Bark was used for the production of many everyday items like floor boards [1], roofs [2], containers [3], as a writing material [4], for cloth [5], for flavoring (cinnamon) [6] and for fishing [7]. Its use as a resource to extract substances probably dates back even to the Pleistocene. Archeological records and recent experiments suggest that Neanderthals have already produced adhesive tar from birch bark [8]. A more prominent, but also more recent example is the tanning of animal skins using the tannins naturally present in tree bark to produce leather, which is considered to be the first manufacturing process of humankind [9]. Bark applications are documented around the European rural periphery until the beginning of the 20th century. Many of these objects can be found in museums all over Europe. For example, in Northern Europe in Finland various household items were made of birch bark and examples can be found eg. in the collection of

**Funding:** CW, FR and ME acknowledge the support of the Cluster of Excellence "Matters of Activity. Image Space Material" funded by the Deutsche Forschungsgemeinschaft (DFG, German Research Foundation) under Germany's Excellence Strategy – EXC 2025-390648296 (https://www.dfg.de). The support of the Max Planck Society is acknowledged by CW, FR, NH and ME. FB was financed by the project "HoBeOpt" (Grant No.: 22008518), which is funded by the Fachagentur Nachwachsende Rohstoffe e.V. (https://www.fnr.de/). The funders had no role in study design, data collection and analysis, decision to publish, or preparation of the manuscript.

**Competing interests:** The authors have declared that no competing interests exist.

the National Museum of Finland (eg museum numbers KB2664b for a woven bag made of birch bark before 1893, KA7424 for shoes made of birch bark before 1906). Another example are Austrian bark huts as seasonal housing for forest workers in the past, as eg shown in the Freilichtmuseum Salzburg.

Until today, the utilization of bark can roughly be classified into three groups: bark used as a bulk material, bark as a resource to extract certain chemical components and bark as an energy source.

In the field of bark extraction, numerous current research projects focus on screening bark for useful components and their efficient extraction for different industries. Prominent examples are birch bark extracts for biomedical applications [10], the development of rigid tannin foams [11, 12] or the usage of bark or its components as adhesives [13, 14]. Regarding the use of the bulk material, research activities concentrate mostly on cork of the mediterranean cork oak (*Quercus suber*, L.) [6]. Recent developments in cork research have shifted from the classical cork-wine utilisation to new cork-based materials and composites which are considered as one of the most promising fields of cork technology [15, 16]. The efforts undertaken to study bark from other tree species for use as a bulk material are limited [6], and only a minor proportion of bark, mainly birch bark, is currently used to create crafts objects such as pots and storage containers [17, 18]. Large quantities of other types of bark are seen as industrial by- or waste-products and used as mulching materials for gardening work or burnt in the wet state directly on site of e.g. sawmills for heat and /or power production despite their lower calorific values and higher ash contents compared to wood [19].

Motivated by adding value to this "by- or waste-product", it has been shown that bark particles can replace wood particles by using higher amounts of adhesives to produce particle boards with acceptable mechanical properties, since an increase in bark content can reduce mechanical performance [20]. Efforts were also made to produce pure bark particle boards [21] and fiberboards. To make use of the orthotropicity of bark, Kain et al. [22] used bark pieces with sizes between 10–30 mm, added 10% urea formaldehyde resin and hot-pressed insulation panels with different densities ranging from 200 kg/m$^3$ to a maximum of 450 kg/m$^3$ and various particle orientations, resulting in a material with directional properties. However, the safe use of formaldehyde-based adhesives is challenging since formaldehyde is classified as carcinogenic [23]. While most bark panels were produced by adding resin, Burrows [24] showed already in 1960 that it is possible to produce resin-free bark particle boards by activating the natural gluing capability of the material by hot-pressing the bark of Douglas fir. Recently, the formaldehyde-binding capacities of bark boards [25] and adhesive-free low-density insulation panels produced with spruce bark were investigated [26].

The examples show that contemporary use of bark is based on using crushed bark pieces and on heavy processing. As a consequence, the inherent structural properties of natural bark are destroyed instead of being used: for living trees, bark has a protective function and covers large areas. One can see it as a natural material, available in large dimensions and quantities. The present work aims to make use of this large-scale raw material and its inherent natural properties such as fibre orientation, anisotropy and gluing potential. To minimize ecological impacts the work focuses on the use of bark of local tree species only. The bark was retrieved in large pieces with the historic method of bark peeling. The goal was to process the retrieved bark pieces into a "standardized" product for potential applications without the incorporation of additional substances that might be opposed to a re-use after the products lifetime.

For this, the raw bark material with different moisture contents was hot-pressed to create pure bark panels without adding any other substance. Structural features, swelling behaviour and mechanical properties of the panels were characterized and discussed with respect to the natural microstructures of the different bark types. The natural availability of a wide range of

different bark types suggests, that bark might not only be a resource for extracting valuable compounds but also a raw material to create new types of purely bio-based panels.

## Material and methods

### Raw material

Freshly felled birch (*Betula pendula*, Roth), larch (*Larix decidua*, Mill.), oak (*Quercus robur*, L.) and pine (*Pinus sylvestris*, L.) were debarked manually in May 2018 in the forest Katharinen-holz, Potsdam, Germany. Most of the tree trunks were cut to ca. 60 cm long segments. After-wards a longitudinal cut was placed in the bark including the bast and cambium (Fig 1). Pieces of bark with lengths between 600–1000 mm and widths of 300–600 mm were removed by hand or with a metal handle (Fig 1). After peeling, the bark pieces were fixed with screws between wood strips to prevent curling upon drying during storage under an outdoor shelter for 6 weeks. After this initial natural drying the wood strips were removed and the flat bark pieces were stored outside under a shelter for at least another 8 weeks until used. Samples with dimensions of 10 cm x 10 cm x bark thickness, 22 cm x 22 cm x bark thickness and 30 cm x 30 cm x bark thickness were cut and stored under laboratory conditions until use.

### Production of flat panels (FP1)

Laboratory-stored pieces of bark with in-plane dimensions of 10 cm x 10 cm, 22 cm x 22 cm and 30 cm x 30 cm were pre-heated at 90 ˚C in an oven without circulating air for 20–30 minutes. Directly after pre-heating two bark pieces were stacked crosswise, with the rhytidome sides facing each other (Fig 2), in a heated hydraulic press (ZSCHOKKE WARTMANN,

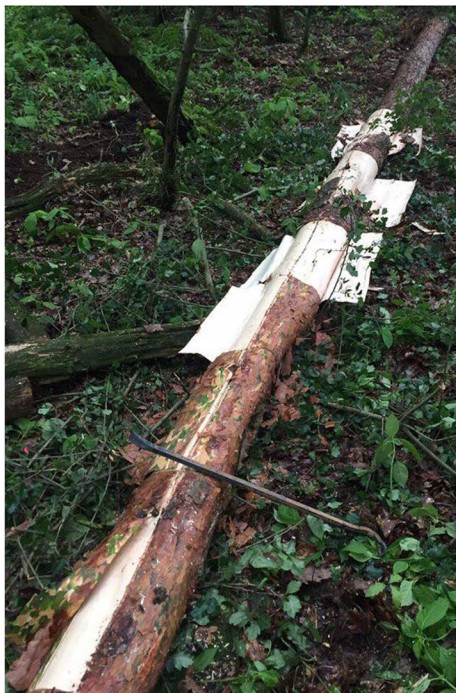 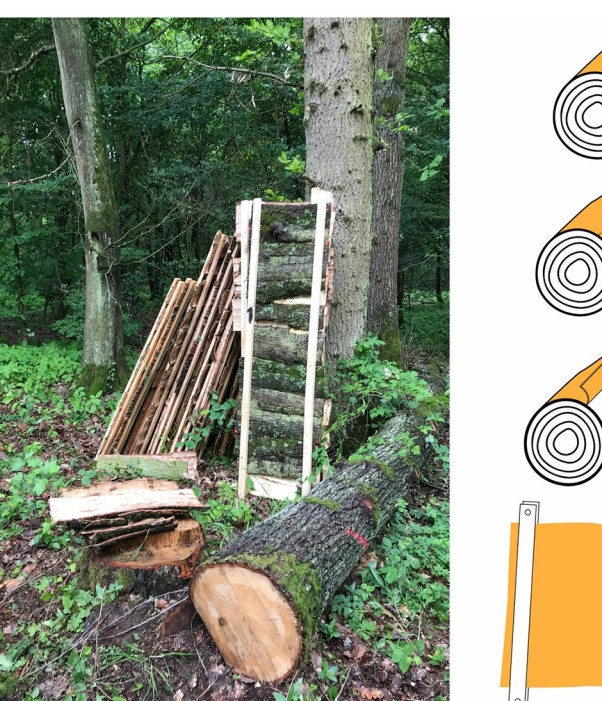 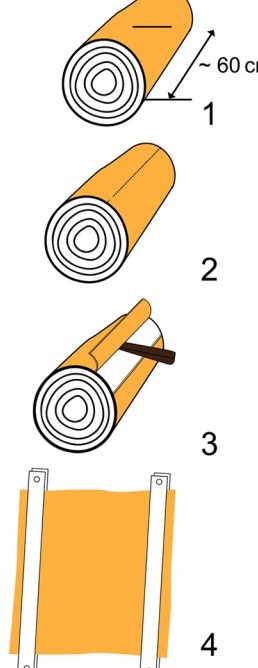

**Fig 1. Process of bark peeling (left), a pile of drying bark fixed between a wood construction (middle) and peeling process (right).** (1) cut of the stem into needed length for bark peeling. (2) by creating a longitudinal cut, the bark including bast and cambium becomes a starting point for the peeling (3) by hand or with a handle, the bark is removed in one piece (4) freshly cut bark is fixed for drying.

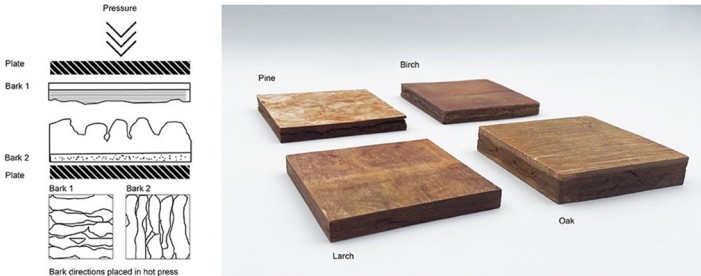

**Fig 2.** Arrangement of the bark pieces in the hot press (left). Produced panels with in-plane dimensions of 10 x 10 cm$^2$ and variable thicknesses determined by the natural structure of the pressed bark pieces (right).

IMEX). To keep the specific pressure constant, the oil pressure was set to 20 bar and 97 bar for 10 cm x 10 cm and 22 cm x 22 cm samples, respectively. The panels were pressed for 20 minutes at 90 ˚C before cooling them down outside the press at room temperature (22˚ C).

## Production of flat panels made of bark with different moisture contents (FP2)

To examine the influence of the moisture content of the raw bark on the panel properties, samples of oak bark and larch bark (both available in large quantities) with four different defined moisture contents were used for panel production in a second panel production series. For both tree species, 16 square bark pieces each (25 x 25 cm$^2$) were stored at 20 ˚C and relative humidities (RH) of 15%, 40%, 65% and 95% until constant weight (less than 0.1% weight change within 24 h) was reached. The bark panels were pressed crosswise, the rhytidome side facing each other, on a Siempelkamp hot press (G. Siempelkamp GmbH & Co., Maschinen- und Anlagenbau, Krefeld, max press area 50 x 100 cm$^2$) with a specific pressure of 97 bar at 90 ˚C and for 25 minutes. Immediately after the pressing process, the panels were removed and allowed to cool in an air-circulated standing position at room temperature for 30 min. This was followed by storage at 20 ˚C and a RH of 65% until equilibrium mass was reached.

## Surface characterization—Surface roughness

The surfaces of bark panels (FP1) were investigated with a digital microscope (Keyence VHX-5000 Microscope) at a magnification of 200x. Surface profiles perpendicular to grain were generated from z-stack images and exported numerically from the software of the microscope. The waviness of the profiles was taken into account by a 2$^{nd}$ derivative spline baseline subtraction over 8 points using the software Origin 2019. For the determination of the average surface roughness ($R_z$), the sampling length of 5 equal and consecutive segments was chosen to 0.25 cm, resulting in a total length of 1.25 cm, according to DIN EN ISO 4288. For each segment, the difference between minimum and maximum profile height was calculated. The average of the values for all 5 segments gave the maximum height of profile for one sample.

## 3-D-microstructure of natural bark and panels

The natural bark material and its structural changes caused by the pressing process (pressure and heat) was characterized by µCT measurements. Native bark samples and bark panels (FP1) were cut with a band saw. The native bark samples of all 4 tree species were cut to the longitudinal-tangential dimensions of approx. 1 x 1 cm$^2$, while their radial dimensions corresponded to the thickness of the bark. The panels of the same tree species were cut to strips with

a width of ca. 1 cm, the cut was 0 and 90 ˚ to the fiber orientation of the pressed bark pieces. The samples were scanned in a micro computed tomograph (RX Solutions, EasyTom150/160) with a microfocus X-ray tube unit (Hamamatsu Photonics K.K. made in Japan) at a tube voltage of 60 kV and 150 µA tube current. The frame rate of the flat panel detector was 2 and the frame averaging was 8. The acquired radiographs of the native bark samples and the panels were reconstructed with the software XAct2 (RX Solutions) and visualized with the software Amira (Thermo Fisher Scientific).

## Density measurements

To assess the degree of densification during the pressing process, the density of both natural bark pieces and the produced panels (FP1) was determined, largely following DIN EN 323. Oak, larch, birch and pine (n = 6 / tree species) bark was cut with a circular saw into 2.5 x 2.5 cm$^2$ large pieces. Additional samples of strongly matured, thick bark of oak and birch were cut to a size of 5 x 5 cm$^2$. The pieces were stored in a climate chamber (VÖTSCH, VCL 4010) at 20 ˚C and 65% RH until constant weight was reached. To account for the rough surface of raw bark samples, their volume was determined with a 3D imaging approach. The conditioned samples were scanned with a micro computed tomograph (EasyTom 150/160 RX Solutions) with a microfocus tube operated at 40 kV and 300 µA in the large focal spot mode. The frame rate of the flat panel imager was 12.5 without averaging, resulting in scan times of approximately 5 minutes. The recorded radiographs (896) were reconstructed with the software XAct. The bark pieces were segmented and their volume was calculated with the software Amira. Immediately after the scan, the samples were stored again in the climate chamber until constant weight was reached. The weight was recorded and the density of each sample was calculated with the previously determined volume. The density of the bark panels was determined on 5 cuboids per species with in-plane dimensions of 2.5 x 2.5 cm$^2$. The panel thickness varied between 5.7 mm and 11.9 mm. The cuboids were stored in a climate chamber at 20 ˚C and 65% RH until their weight changed less than 0.1% within 24 hours. The sample dimensions were determined with a caliper directly after weighing.

## Swelling and shrinkage of native bark cubes

Materials in applications are continuously exposed to changing environmental conditions (relative humidity and temperature) and water contact is one of the big challenges encountered when using lignocellulosic materials not only due to the elevated risk of fungal decay, but as it strongly influences the material properties, too. While directional swelling movements of wood and derived timber products are comparably well described [27, 28], only little is known about swelling and shrinkage of bark.

To compare the swelling and shrinkage behavior of the native bark and the compressed panels, bark cubes without cracks of oak and larch were produced. In a first step, small sized strips were cut from dried bark. For each species two subseries of specimens (oak n = 12, larch n = 10) were produced originating from two different trees. The strips were planed square on two adjacent sides (radial/tangential plane) with a small hand plane. Afterwards both opposite sides were trimmed with a microtome (Leica RM 2255) to a thickness of 10 mm each. The resulting square strips were cut to a length of 10 mm with an electric mitre saw. Areas which included cracks or visible voids were discarded. The resulting bark cubes were conditioned in a climate chamber (Vötsch VCL 4010) at 65%, 40%, 20%, 85%, and 0% relative humidity at 20 ˚C. Throughout the experiment the climatic conditions were measured with a calibrated sensor (SHT85, Sensirion). The lengths of the samples along their three anatomical directions were measured with an outside micrometer at constant weight (balance: PCE AB100C) which

was usually reached after 3–4 days of conditioning. The dry state of 0% relative humidity was achieved by a direct input of compressed air into the chamber (dew point below -30 ˚C). For the evaluation, the results of the moistening state at 65% relative humidity served as reference.

## Swelling and shrinkage properties of larch and oak panels pressed at different moisture contents (FP2)

To investigate the sorption, swelling and shrinkage properties of the panels (FP2), larch and oak samples (55 mm x 55 mm x panel thickness, n = 10 / species) were exposed to five sequential climatic conditions. A climate chamber (Feutron type "035/09") was used for conditioning. Exposure to one climatic condition was completed when all samples had reached an equilibrium state. The dimensions of the panels were measured with a digital sliding caliper (accuracy of ± 0,01 mm), changes in thickness and length were analyzed. The equilibrium sample dimensions of the starting condition (20 ˚C, 65% RH) were defined as initial point for the calculations of the percentage swelling and shrinkage dimensions, according to standard DIN EN 317. To investigate the shrinkage of the panels, the relative humidity was reduced from 65% to 40% RH and 20% RH (always at 20 ˚C). This was followed by an increase in relative humidity to 85% at 20 ˚C. Finally, the samples were dried at 80 ˚C and laboratory RH.

## 3-point bending tests

To investigate the mechanical properties of the bark panels (FP1), samples with a width of 2.5 cm and a length of 21 cm were cut for 3-point bending tests. Care was taken to cut at 0 ˚ and 90 ˚ angles to the grain of the panels. The samples were then stored until testing at 20 ˚C and 65% RH until constant weight was reached. The mean thickness and mean width of each sample was calculated from three measurements, taken with a caliper at the center and on both support points. Bending tests were conducted on a Zwick universal testing machine equipped with a load cell with a maximum capacity of 10 kN. Two test series were performed for each tree species with the grain orientation either along or across the tension side of the bending sample. The test span was 15 cm and the test speed was set at 0.1 mm s$^{-1}$. The samples (oak$_{long.}$ n = 7, oak$_{perp.}$ n = 6, larch$_{long.}$ n = 8, larch$_{perp.}$ n = 11) were tested until fracture, deflection was measured via cross-head displacement. The determination of the bending modulus E was calculated according to Eq 1:

$$E = \frac{\Delta F * l_1^3}{4 * b * t^3 * \Delta \varepsilon} \tag{1}$$

where $\Delta F$ is the force increase in the linear region, $l_1$ the distance between the supports, b the sample width, t the sample height and $\Delta \varepsilon$ the deformation increases in the middle of the beam.

## Determination of transverse tensile strength (FP2)

To characterize the effect of moisture content during pressing on the connection between the crosswise pressed bark pieces, transverse tensile strength perpendicular to the plane of panels was determined according to the standard DIN EN 219. The conditioned bark panels were cut to samples with a size of 50 mm × 50 mm x panel thickness. Ten samples cut out of two panels of one tree species and pressed with the defined moisture content and conditions were mounted with polyurethane adhesive (Kleiberit PUR 501) between beech plywood yokes. The glued specimens were clamped into prefabricated templates using screw jaws. Additional wooden spacers prevented the specimens from slipping sideways between the yokes. The transverse tensile tests were carried out on an universal tensile machine (1484, Zwick Roell GmbH) equipped with a load cell with a maximum capacity of 200 kN. Prior to material

testing, the densities and equilibrium moisture contents were determined. The transverse tensile strength was calculated according to Eq 2:

$$\sigma_Z = \frac{F_{max}}{a * b} \tag{2}$$

Where $\sigma_Z$ is the transverse tensile strength, $F_{max}$ the maximum force and $a*b$ the cross sectional area.

The results of the panels manufactured of bark with different moisture contents were compared with each other and statistically analyzed with the Mann-Whitney test (OriginPro 2021b).

## Production of curved panels

In a second approach, the feasibility to create 3D shaped geometries was tested. Larch, pine, oak and birch bark without glue or resin (Fig 9) were molded with the goal to explore limits in achievable curvatures. Therefore, bark with in-plane dimensions of 10 cm x 10 cm was preheated at 90 ˚C. Directly after pre-heating the bark piece was placed in curved molds with the axial bark direction following the curve. Molds with different inner and outer diameters (10 mm / 20 mm; 20 mm / 30 mm; 30 mm / 40 mm; 40 mm / 50 mm) as shown in Fig 9 were used. The mold was placed in a heated hydraulic press (ZSCHOKKE WARTMANN, IMEX). For 10 x 10 cm$^2$ samples the oil pressure was set between 3 and 20 bar—depending on the dimensions of the mold and the piece of bark. The samples were pressed for 20 minutes at 90 ˚C. Afterwards the curved panels were removed from the mold for cooling down at room temperature (22 ˚C).

## Results and discussion

### Production process

The manual peeling process of locally harvested bark provided initial, qualitative information about the peeling properties: birch, oak, larch and pine manual peeling in spring is feasible, indicating that it can be easily processed by human power for small amounts of trees. However, it is documented that manual debarking of Scots pine is heavy physical work [29]. Since bark was peeled circumferential to the stem axis, it is conceivable that industrial processes, similar to the ones in veneer production, could be adapted especially for large-scale bark harvests. It has been shown that the growing season can remarkably influence the adhesion between wood and bark [30] indicating that debarking during the growing season demands less effort.

The first tests to produce flat panels out of laboratory-stored and pre-heated pieces of bark (FP1) were successful for all pieces of bark of the four tree species and resulted in panels with species-characteristic textures and colors (Fig 2). The experiments with bark stored at defined relative humidities (FP2) revealed, that bark stored at a RH > 95% could not be pressed to panels since the high amount of water impeded a proper bonding while squeezed out of the samples. Furthermore, storage at 95% RH promoted growth of microorganisms on the cambial side of the bark. This might be attributed to the availability of easily accessible nutrients for microbes (e.g. sugars).

The panels can be easily manufactured by milling or cutting. Panels, as shown in Fig 2, could be cut simply with a circular saw, resulting in a smooth surface of the edges, where the boundary lines between the phloem (living during harvest) and the rhytidome (dead bark) become clearly visible.

## Surface characterization of raw bark and panels

Since in-plane surfaces appeared smooth after the pressing process, their roughness was determined 90° to the grain on the raw panels and gave $R_z$ values of 33.9 μm for larch, 15.1 μm for pine, 13.2 μm for oak and 21.6 μm for birch. These roughnesses are comparable with sanded surfaces of solid wood [31]. Considering their natural appearances (Fig 2) and the machinability, applications for furniture and paneling without extensive surface treatment can be suggested.

## Structure and density of bark and bark panels

The effects of the pressing process on the bulk material are visualized based on μCT scans of native bark and panels of all four tree species (Fig 3). The flake-like native bark types of pine

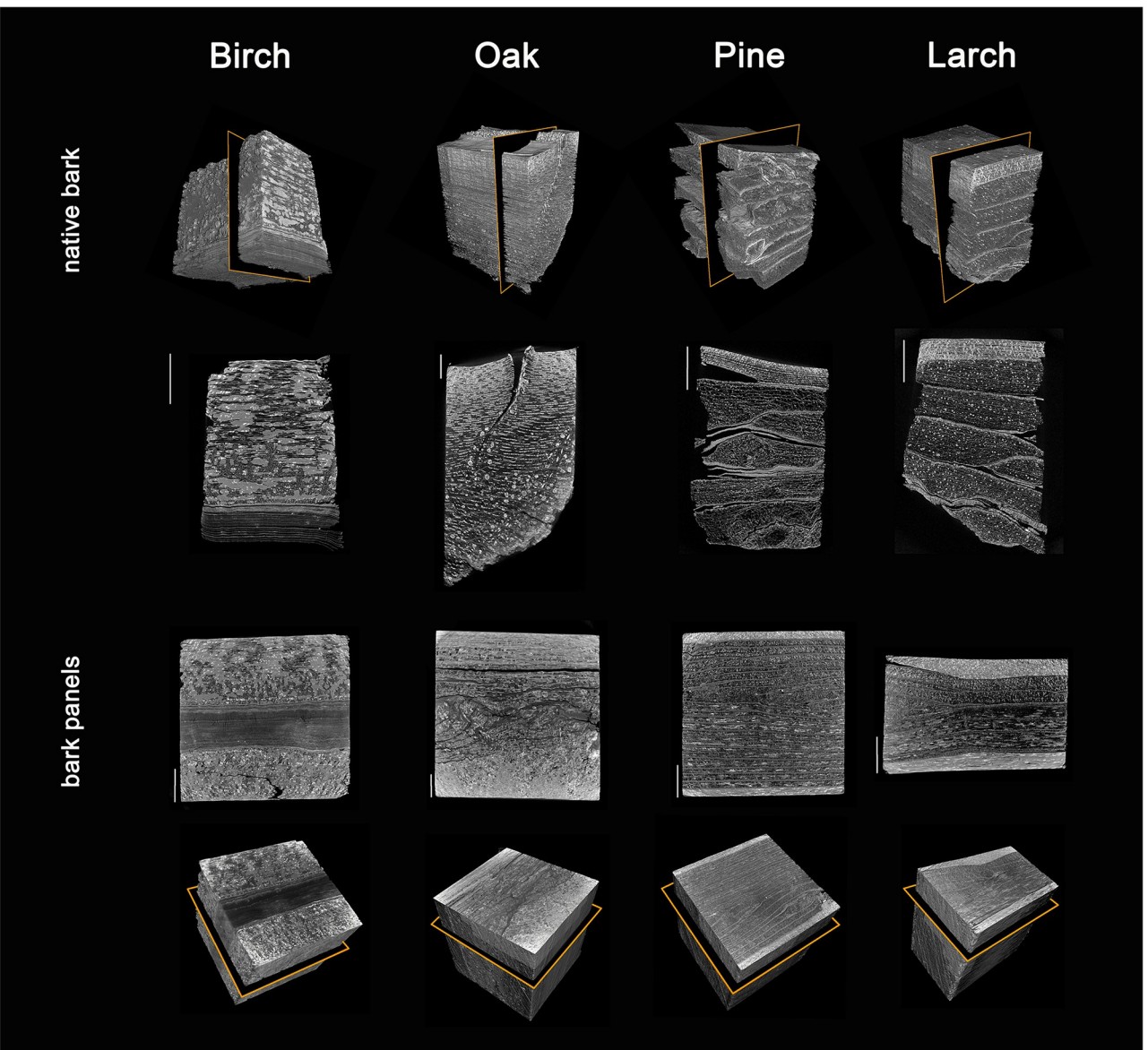

**Fig 3. μCT scans of native bark pieces and compressed bark.** Black slices in rectangular volumes indicate the virtual cutting directions of the slices next to the volumes. Bars 2 mm.

and larch possess numerous cracks, mainly in the periderm or at the interface between periderm and dead phloem. These cracks disappear upon pressing. In addition, the pressing process leads to a strong densification of the sieve cells and dilated parenchyma of the phloem (both young and old), whereas the sclereids of larch phloem and many phellogen cells of larch and pine retain their shapes. In comparison, the microstructures of the native hardwood bark pieces of oak and birch are fundamentally different to the two described softwood species. The fiberless birch bark is characterized by groups of sclereids in the phloem, the phellem consists of the prominent papery layers. In oak bark numerous fiber bands are present and sclereids are found in groups. The compression process for panel formation results in a collapse of sieve tubes and parenchyma cells, and the distance between fiber bands decreases considerably. In birch, on the other hand, the irregularly distributed groups of sclereids seem to prevent the thin-walled elements (sieve tubes and parenchyma) from further compaction.

The described anatomical differences are also reflected in the densities of the native bark and the panels, determined at 20 ˚C and 65% RH (Fig 4). Due to their structural composition, densities of bark samples increase differently upon pressing. Native birch bark has a high

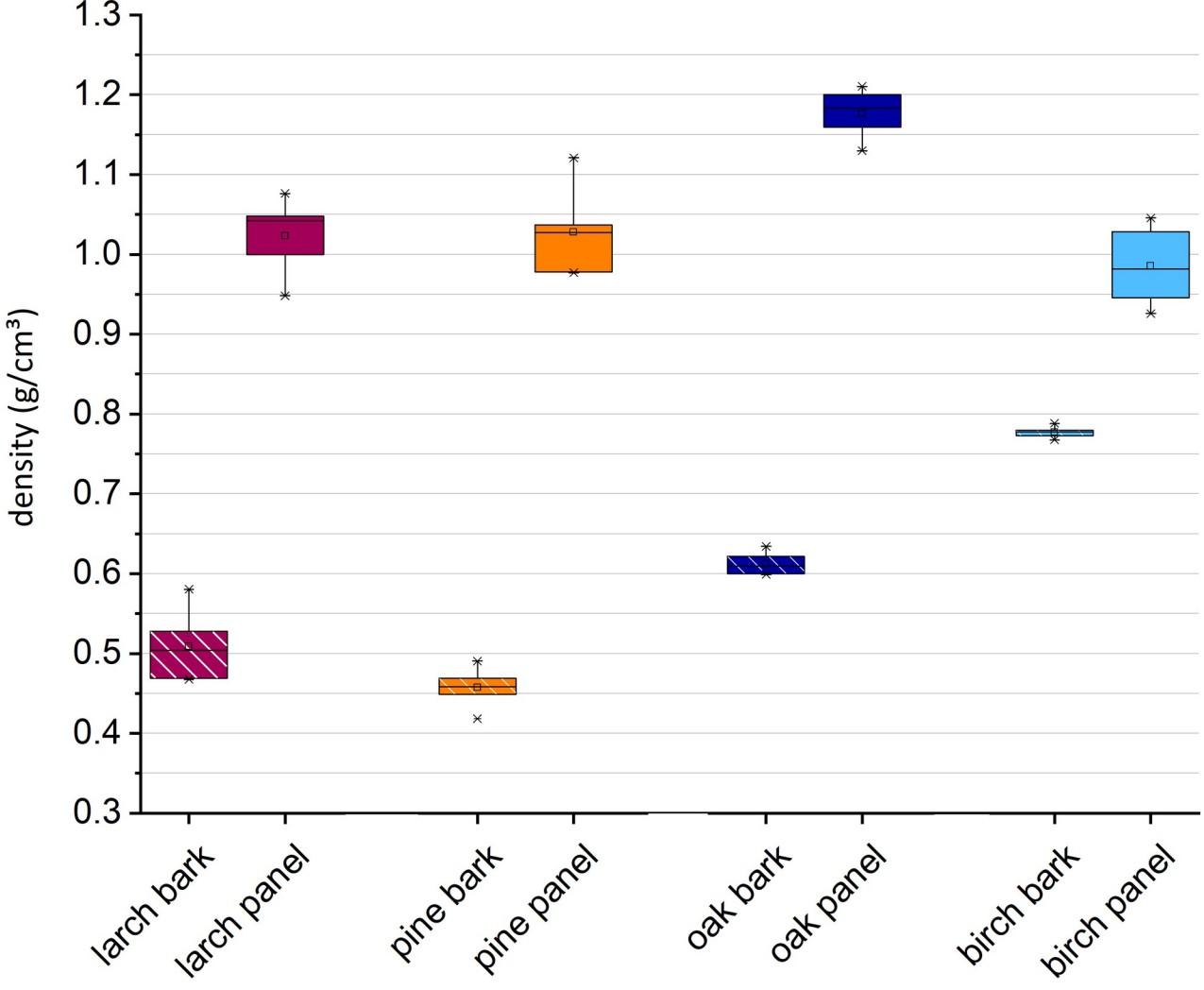

**Fig 4. Densities of native bark pieces and compressed bark (FP1).**

density between ~ 0.78 g/cm$^3$ for thick bark pieces (Fig 4) and ~ 0.83 g/cm$^3$ for thin bark pieces (data in repository), which can be explained by the high amount of dense sclereid groups. The pressing procedure resulted in a slight increase in density (0.99 g/cm$^3$). A possible explanation is that the maximal compression is reached when sclereid groups get in contact with each other. Oak bark was less dense than birch with a mean density of 0.71 g/cm$^3$ (thin pieces, data in repository) and 0.61 g/cm$^3$ for thick pieces as shown in Fig 4. The compression process led to panels with a density as high as 1.18 g/cm$^3$. Native larch bark had a density of 0.51 g/cm$^3$ and pine of 0.46 g/cm$^3$, the same pressing parameters led to panels with densities of 1.03 +/- 0.06 g/cm$^3$ (pine) and 1.02 +/- 0.05 g/cm$^3$ (larch). These results indicate that native birch bark is more resistant to lateral forces, compared to the other tree species.

## Mechanical characterisation

For an initial evaluation of the mechanical properties of bark panels (FP1), 3-point-bending tests were performed on larch and oak bark panels. Limited availability of raw material did not allow tests on bark panels of the other tree species. Since the sample geometries deviated from standard tests—again caused by limited availability of material—particle board samples with a density of ~ 0.7 g/cm$^3$ and the same sample geometries as the bark panels were tested in the same manner and are shown as gray lines in Fig 5b and 5c to allow a comparison with a well-known wood-based material.

The 3-point-bending tests showed a strong effect of the fiber orientation on the modulus (Fig 5b) and on the maximum bending stress (Fig 5c). This effect is more pronounced for oak and can possibly be attributed to the presence of numerous fiber bundles in the phloem. Such strong effects of the fiber orientation on the tension side of bending beams are not surprising and also known from other wood-based materials, e.g. veneer-based wood products [32]. The calculated bending moduli of the bark panels (Fig 6) are in the same order of magnitude as reported values from wood-based panels such as fiber or particle boards [33] and are considerably lower than those of larch or oak solid wood which often exceed 10 GPa [33]. The consideration of the different panel densities (oak ~1.2 g/cm$^3$, larch ~ 1 g/cm$^3$) leads to less pronounced differences in specific properties between the different types of panels [26]. However, the bending properties of oak panels with longitudinal fiber orientation at the bottom layer are superior compared to all other panels. This first attempt to create pure bark panels and the initial bending experiments suggest promising properties for binderless bark panels. Optimization steps of the processing conditions (temperature, pressure and water content of the raw material) were not performed at this stage but seem promising.

In the following, we started to initially investigate the role of moisture during processing on the transverse panel properties by pressing raw bark pieces with different moisture contents to panels (FP2), (Table 1). The evaluation of the connection between the two bark pieces of a panel (FP2) was based on transverse tensile tests (Fig 6) and swelling experiments (Fig 7).

The transverse tensile strength of the oak bark panels P3 (raw material stored at 20 ˚C and 65% RH) is more than twice as high as for the panels P1 (20 ˚C and 15% RH). Statistical analysis showed no significant differences between 20 ˚C / 40% RH and 20˚C / 65% RH. For the larch panels, there was no clear effect of the moisture content of the raw material on the transverse tensile strength which was generally lower compared to the oak bark panels. A possible explanation for the low transverse tensile strength of larch could be its bark structure: natural larch bark appears flake-like and contains numerous "weak spots" and cracks. In the panels they might serve as starting points for crack initiation, crack growth and fracture. In the future, a particular focus should be laid on the bonding of the flakes within larch bark pieces, as well as on the connection between two pieces. In contrast, the water content of oak bark during

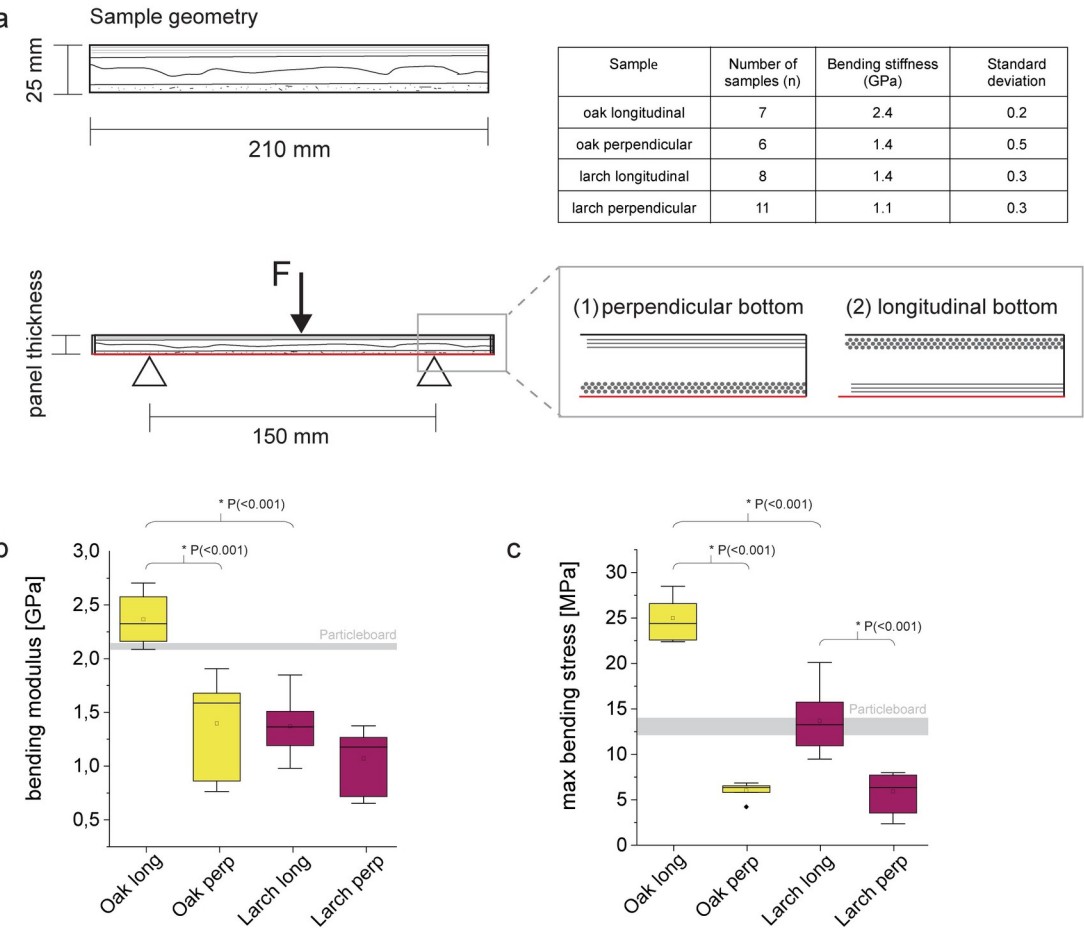

**Fig 5.** Mechanical data of three-point-bending experiments (a) samples with a length of 210 mm and a width of 25 mm were cut and tested with a test span of 150 mm. Sample thickness corresponded with panel thickness, the orientation of the fibers at the bottom was either perpendicular (1) or longitudinal (2). (b) bending modulus of the tested samples (c) max bending stress of the specimens. Boxes show 25th–75th percentile, small rectangle in box is the mean and the line the median of the samples, stars correspond to outliers. Gray line in the diagrams shows bending modulus of the tested particle board with a density of ~ 0.7 g/cm³ (line thickness equals interquartile range from 25th– 5th percentile).

pressing affected the bonding between and possibly, but less likely, within the bark pieces: a water-induced reduction in hardness of the raw material, similar to wood [34], might lead to a deeper indentation of the serrated bark pieces into each other and hence to a stronger mechanical connection. The formation of hydrogen bonds might also play a role as well as a faster heat transport due to a higher moisture content [35].

The achieved transverse tensile strengths of oak (P3, Fig 6) is comparable to the transverse tensile strengths of MDF boards (0.55 N/mm²), a UF resin-bonded particleboard (0.65 N/mm²) and a UF resin-bonded particleboard with 40% spruce bark (0.58 N/mm²) [35].

## Effects of water on swelling

**Swelling of native bark.** To evaluate the hygroscopic behavior of bark and bark panels, swelling properties of the native bark samples were determined in the different anatomical directions on undamaged isolated pieces of material.

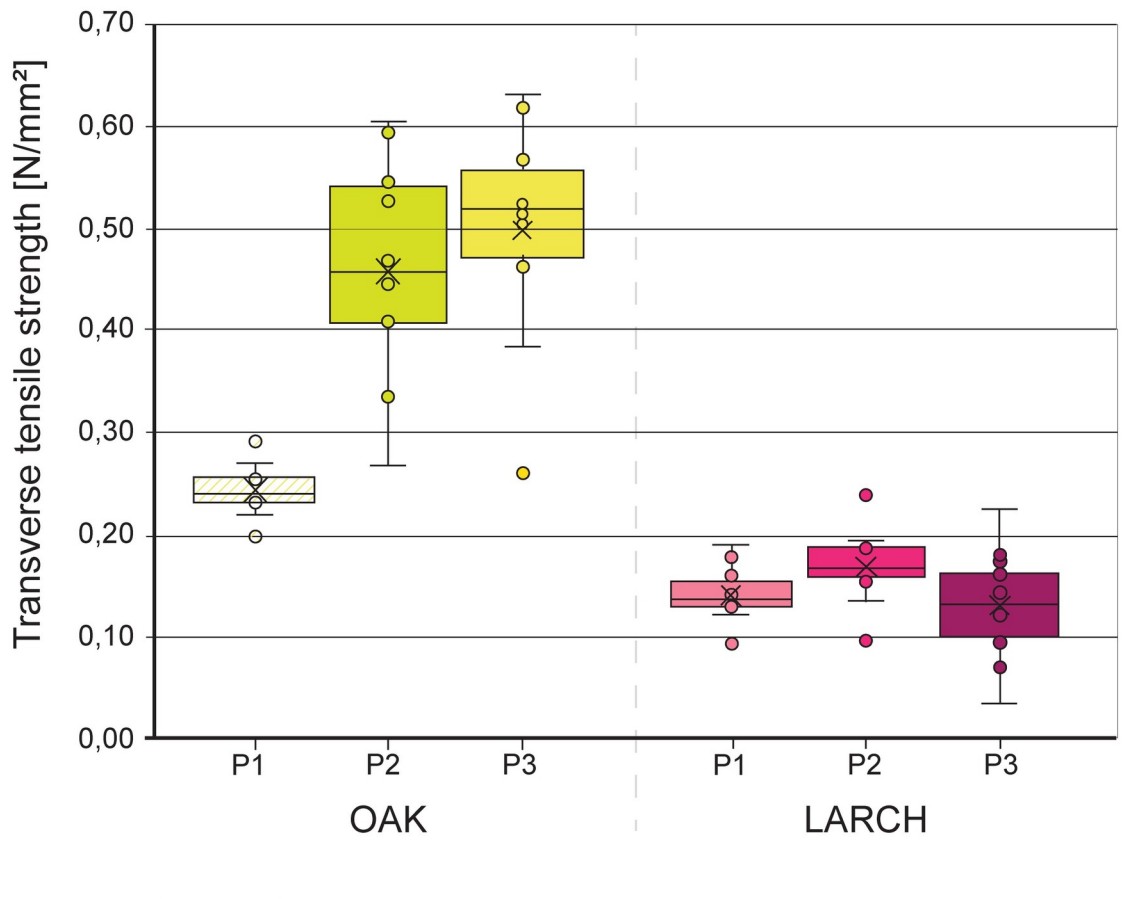

**Fig 6. Transverse tensile strengths of the bark panels divided into the press moisture groups P1, P2 and P3 of both bark types.** Boxplots with median as middle line and arithmetic mean as cross. The whisker length is a maximum of 1.5 times the interquartile range.

**Table 1. Density, equlibirum moisture content, and transverse tensile strength of panels made of bark pressed with different moisture contents and afterwards stored at 65% RH and 20 ˚C until constant weigth.**

| Storage conditions raw material [˚C/% RH] | Bark sample | Transverse tensile strength [MPa] | Density [g/cm³] | Equilibrium moisture content $u_{65}$ [%] |
|---|---|---|---|---|
| P1 (20 / 15) | Oak | 0,24 ± 0,02 | 1,11 ± 0,03 | 8,12 ± 0,19 |
| | Larch | 0,14 ± 0,03 | 1,00 ± 0,03 | 8,71 ± 0,27 |
| P2 (20 / 40) | Oak | 0,46 ± 0,10 | 1,13 ± 0,05 | 8,95 ± 0,25 |
| | Larch | 0,17 ± 0,04 | 1,04 ± 0,07 | 10,08 ± 0,40 |
| P 3 (20 / 65) | Oak | 0,50 ± 0,10 | 1,22 ± 0,22 | 11,02 ± 0,53 |
| | Larch | 0,13 ± 0,05 | 0,95 ± 0,09 | 12,90 ± 0,30 |

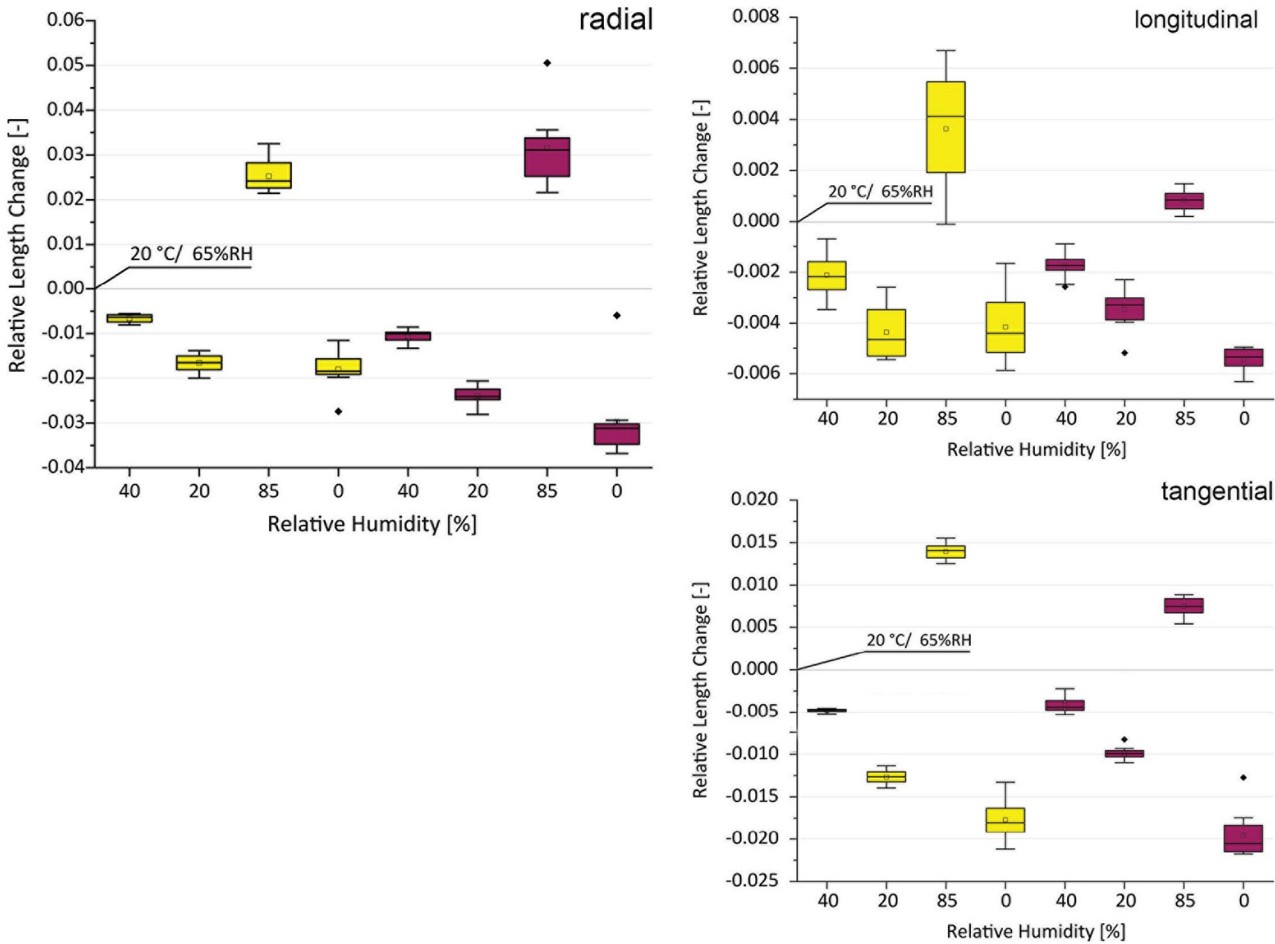

**Fig 7. Swelling and shrinkage behavior of native oak (yellow) and native larch (purple) upon changing humidity levels.** All levels show the thickness swelling compared to the original sample dimension at RH 65% [0].

The dimensions of the samples stored at 20 ˚C and 65% RH were defined as the initial lengths [0]. The most pronounced swelling and shrinkage of the bark cubes was measured in radial direction for larch and oak. Upon drying the samples, oak and larch shrank ~ 2% and ~ 3% respectively. Increasing the relative humidity to 85% led to swelling of oak and larch by ~ 2.3% and ~ 3.3%. In tangential direction samples shrank ~ 2% (oak) and ~ 2.2% (larch) and swelled up to ~ 1.4% (oak) and ~ 0.7% (larch). In the longitudinal direction, the swelling is much less pronounced.

The present and previously reported work [36] found the most pronounced swelling in the transverse directions which include tangential and radial directions. Surprisingly, free swelling seems to be more pronounced in the radial than in the tangential direction, however it can also be the other way round as shown in Raczkowskis work [37]. At this stage one can only speculate about the reasons, but supposedly the irregular distribution of cracks and the position of flakes, as well as their curvatures which do not necessarily follow the curvature of the xylem growth rings, might play a role for the somewhat unpredictable behavior. Nevertheless, the most pronounced swelling takes place in one of the transverse directions which coincides with the direction of compression. Slightly different swelling values of our work compared to

Raczkowki (1979) can be explained by different humidity levels (85% RH here and water immersed in [36]).

**Swelling of panels.** The panels made of oak and larch with different moisture contents (P1, P2 and P3) showed clear differences in swelling behavior when sequentially exposed to different climates (temperature and RH). The dimensions after storage at 20 °C and 65% RH were defined as initial (Fig 8). Upon reducing the RH to 40% and then to 20%, the panel

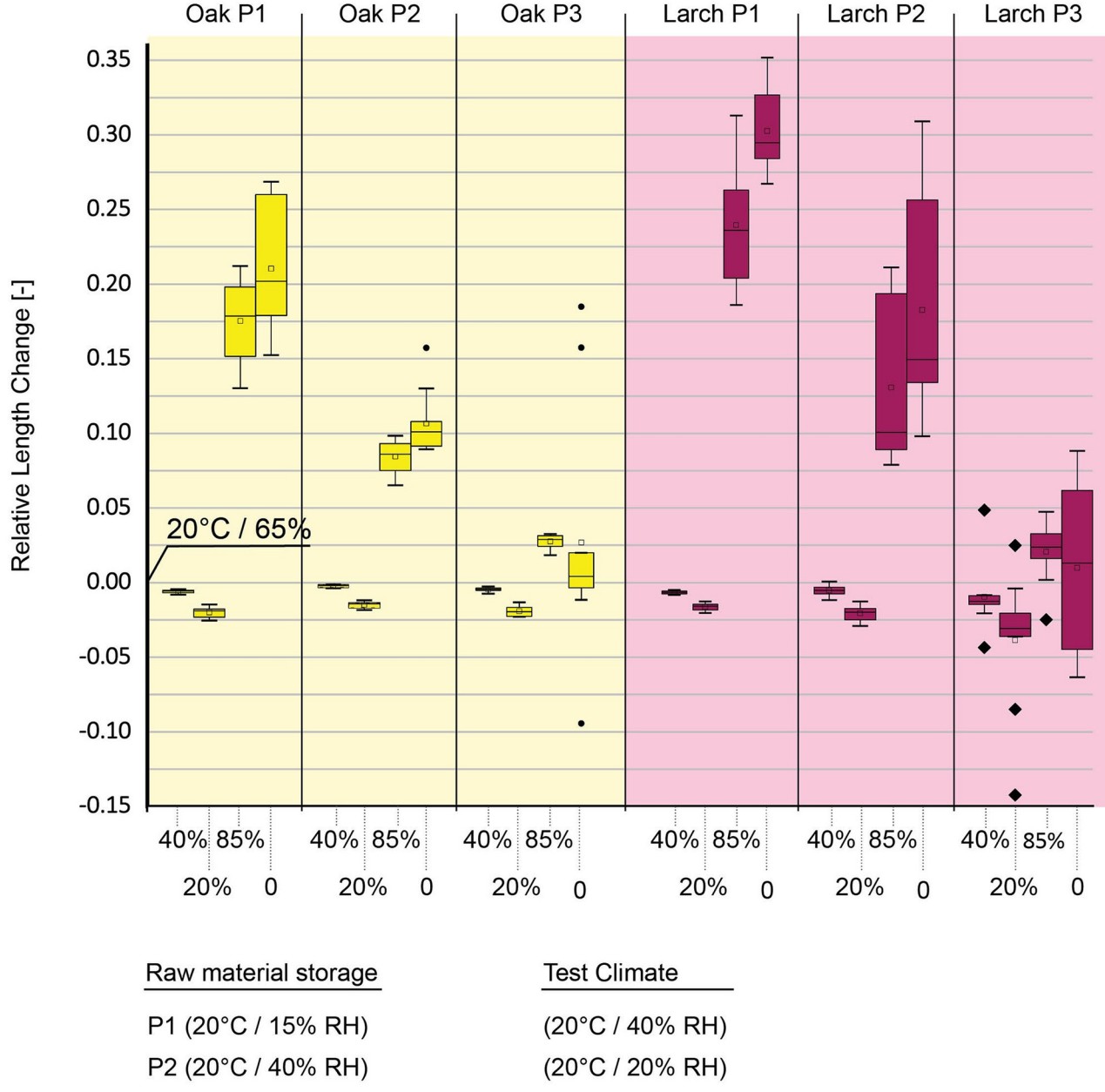

**Fig 8. Swelling and shrinkage behavior of oak (yellow) and larch (pink) panels in the climate test.**

thicknesses decreased for all types of bark pressed with different moisture contents (15% RH, 40% RH, 65% RH). Increasing humidity to 85% RH led to pronounced thickness swelling. Large differences were observed between panels fabricated with different moisture contents of the raw material. Raw material equilibrated at 65% RH prior to hot-pressing (20 ˚C, 65% RH) showed a higher dimensional stability compared to those containing smaller amounts of water (20 ˚C, 15% RH and 20 ˚C, 40% RH). While the panels fabricated of bark stored at low RHs (20% and 40%) showed large swelling (20–30%) upon exposure to 80% RH, the swelling of the panels pressed after storage at 65% RH was in a similar range as the swelling of native bark cubes (~ 1–4%) (Figs 7 and 8). This is a strong indication of only a small spring-back effect at this moisture content induced by the heat- and pressure-exposure and the important role of water presence in the raw material. The exact mechanism, however, remains unknown and further studies are required.

Surprisingly, subsequent drying at 80 ˚C did not lead to shrinkage in all panels. It is possible that structural defects during swelling and/or drying occurred leading to cracks within the boards, which prevent recovering the initial state.

### 3D shaping of bark

To evaluate the potential of molding bark into more complex forms, bark was pressed into different 3D shapes. The preliminary tests showed that oak bark can be bent to radii of 40–50 mm along the fiber direction without visible damage of the raw material (Fig 9).

Radii between 40 and 20mm are possible, although with a certain fiber relocation. Structures with radii below 20 mm could not be realized. The approach was not successful for birch, larch and pine. Birch developed random cracks in all directions, which might be explained by the presence of sclereid clusters and absence of fibers (Fig 3). The softwood bark pieces of larch and pine could be shaped into stable 3D geometries with radii from 20–50 mm, however,

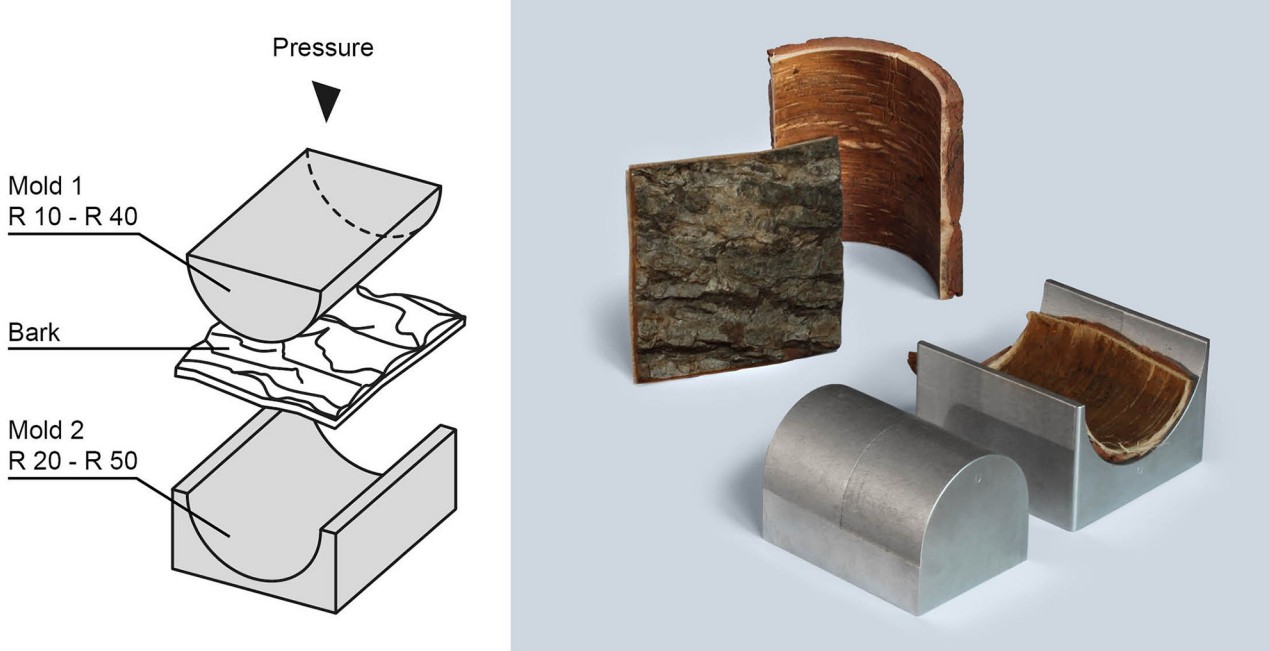

**Fig 9.** Illustration of the pressing process (left). Produced curves with a radius 50mm and a random thickness caused by the natural structure of the pressed (oak) bark piece (right).

significant damage of the structure occurred. It is conceivable that the stability is based on inherent gluing abilities, e.g. resins, tannins, becoming activated by the applied pressure and heat [37]. In summary, these initial findings show, that only bark with long fibers (e.g. oak) can be deformed three-dimensionally without visible damage. The bark molding approach provides a possibility to create 3D shapes without the need for CNC fabrication techniques and consequently without creation of additional waste by milling.

## Conclusions

This proof of concept paper shows that the production of adhesive-free bark panels is possible by a densification process. Bending modulus and strength as well as the transverse tensile strength similar to wood-based-panels such as particle boards were achieved. Experiments with adjusted water content of the raw material showed that bark moisture during the pressure process affects the swelling behavior and also mechanical properties. The transverse swelling of the panels when exposed to 80% RH ranges from ~ 3% up to 30%, depending on the processing conditions. Partly, this swelling remains permanent, even upon drying. The mechanisms underlying the tight connection of the bark pieces remain unclear and require further research as well as new approaches to improve the dimensional stability. Effects of process pressure and temperature on the panel properties have not been studied so far. However, optimized process parameters bear the potential to increase the bonding between the bark pieces.

This work highlights that biogenic resources, often considered as waste, can be processed into products without incorporating additional substances. A major advantage of "pure" one-component products is that no separation of components after their life-time is required. Even though the structure of the raw material is altered, the basic building blocks remain the same and hence can still be easily used for subsequent processing such as extraction of chemicals or fibers, or as a fuel for energy production. Especially through the subsequent possibility of chemical component extraction a cascade use of the material is enabled, which maximizes resource efficiency. The bark panels and molded 3D shapes may serve for applications without water contact (e.g. interior design). Particularly the surface smoothness of panels comparable with sanded wood surfaces (without any further surface treatment) facilitates applications in furniture as well as in the field of packaging.

## Acknowledgments

We wish to thank the forester Uwe Peschke for providing trees for the peeling process. We also thank Jan von Szada-Borryszkowski, Marco Bott, Tobias Schmidt and Florian Weisz for their support during peeling.

## Author Contributions

**Conceptualization:** Charlett Wenig, Nils Horbelt.

**Data curation:** Michaela Eder.

**Formal analysis:** Charlett Wenig, Friedrich Reppe, Nils Horbelt.

**Funding acquisition:** Michaela Eder.

**Investigation:** Friedrich Reppe, Nils Horbelt, Jaromir Spener, Marion Frey, Michaela Eder.

**Methodology:** Charlett Wenig, Friedrich Reppe, Nils Horbelt, Jaromir Spener, Michaela Eder.

**Project administration:** Michaela Eder.

**Resources:** Ingo Burgert.

**Supervision:** Ferréol Berendt, Tobias Cremer, Ingo Burgert, Michaela Eder.

**Validation:** Charlett Wenig.

**Visualization:** Charlett Wenig, Friedrich Reppe, Nils Horbelt.

**Writing – original draft:** Charlett Wenig, Michaela Eder.

**Writing – review & editing:** Charlett Wenig, Friedrich Reppe, Nils Horbelt, Ferréol Berendt, Tobias Cremer, Marion Frey, Ingo Burgert, Michaela Eder.

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
