## [Decision Letter · Decision Letter 0]

10 Oct 2022

PONE-D-22-24770Adhesives free bark panels: an alternative application for a waste materialPLOS ONE

Dear Dr. Eder,

Thank you for submitting your manuscript to PLOS ONE. After careful consideration, we feel that it has merit but does not fully meet PLOS ONE’s publication criteria as it currently stands. Therefore, we invite you to submit a revised version of the manuscript that addresses the points raised during the review process.

We look forward to receiving your revised manuscript.

Kind regards,

Yasir Nawab, PhD

Academic Editor

PLOS ONE

Reviewers' comments:

Reviewer's Responses to Questions

**Comments to the Author**

1. Is the manuscript technically sound, and do the data support the conclusions?

Reviewer #1: Yes

Reviewer #2: Yes

2. Has the statistical analysis been performed appropriately and rigorously? 

Reviewer #1: Yes

Reviewer #2: Yes

3. Have the authors made all data underlying the findings in their manuscript fully available?

Reviewer #1: Yes

Reviewer #2: Yes

4. Is the manuscript presented in an intelligible fashion and written in standard English?

Reviewer #1: Yes

Reviewer #2: Yes

5. Review Comments to the Author

Reviewer #1: Wenig et al provide a nice comperehensive study in which the properties of adhesive free bark panels from bark of different origin are investigated. The study is relevant in terms of the use of a waste material in bulk material application avoiding potentially toxic adhesives that are commonly used. The findings are clearly presented and the conclusions are sound. Publication in PLOS ONE is recommended and only some minor recommendations for improvements are provided:

Introduction:

-The author refer to studies performed in the mid-20th century but then give citations for manuscript in the 21st century.

- When refering to the study by Burrows et al it seems relevant to mention the type of bar used (Douglas Fir).

Results and discussion:

- It took me a bit to figure out what the two pictures on the bottom of Figure 2a meant. I recommend adding a label

- Also I recommend to add enlargement of the side view for the panels to clearly show that the boundry lines are visible (now this only is the case for Oak)

- Figure 3 seems to be placed in the wrong section

- Table 1 is not refered to in the section it is placed in

- Table 1 column 4 should be density

References:

- Please check formatting, some could not be easily traced.

- What is the footnote at the end of the references for?

Reviewer #2: The work is well written and relevant. Correct methodology.The results were analyzed correctly. Some recommendations: objetives at the end of the introduction and put specific objetives. Methodology: cite the age, diameter and height of the trees. Reason for choosing species. Results and discussions: to discuss the relationship of the influence of tree diameter on debarking. Mention the thickness of the barks. Amount of phloem and periderm. Show the cells of the barks.

6. PLOS authors have the option to publish the peer review history of their article (what does this mean?). If published, this will include your full peer review and any attached files.

Reviewer #1: No

Reviewer #2: No

---

## [Author Response · Author response to Decision Letter 0]

28 Nov 2022

Dear Editors, 

First of all we wish to thank you for your time and efforts. In the following we are answering to the additional requirements. The answers to the reviewer comments are below.

Sincerely yours,

Charlett Wenig and Michaela Eder on behalf of all authors

Thank you for sending the instructions. We followed the guidelines.

We will activate the link, created in edmond, upon acceptance https://doi.org/10.17617/3.AZRWF3 and all the data will be accessible there. The µCT data will also be available in a compressed form. 

The ORCID iD has now been validated in the Editorial manager.

Since the data is not shown in Figure 4 but accessible in the data repository, we changed to text to “data in repository”.

We reviewed the reference list (changes not marked) and replaced 2 references by better fitting ones (in red)

We also wanted to mention that Figure 2 and Figure 7 have been modified.

Response to Reviewers' comments:

Dear Reviewers,

First of all we wish to thank you for your their time and efforts to read and carefully review our manuscript. In the following we will respond to raised questions, concerns and comments point by point. The changes in the manuscript are in red, a few corrected typos are not specifically marked.

Sincerely yours,

Charlett Wenig and Michaela Eder on behalf of all authors

Reviewer #1: Wenig et al provide a nice comperehensive study in which the properties of adhesive free bark panels from bark of different origin are investigated. The study is relevant in terms of the use of a waste material in bulk material application avoiding potentially toxic adhesives that are commonly used. The findings are clearly presented and the conclusions are sound. Publication in PLOS ONE is recommended and only some minor recommendations for improvements are provided:

Introduction:

-The author refer to studies performed in the mid-20th century but then give citations for manuscript in the 21st century. 

Thank you for pointing this out! We removed the text relating to the 20th century since the more recent studies are more relevant.

- When refering to the study by Burrows et al it seems relevant to mention the type of bar used (Douglas Fir). 

We agree that the bark type is important and added the information 

Results and discussion:

- It took me a bit to figure out what the two pictures on the bottom of Figure 2a meant. I recommend adding a label 

we added a label and hope that it is clearer now

- Also I recommend to add enlargement of the side view for the panels to clearly show that the boundry lines are visible (now this only is the case for Oak) 

the side view is now enlarged

- Figure 3 seems to be placed in the wrong section 

we moved it to the appropriate section

- Table 1 is not refered to in the section it is placed in 

we changed this

- Table 1 column 4 should be density 

thank you, we corrected the column heading

References:

- Please check formatting, some could not be easily traced. 

We checked formatting

- What is the footnote at the end of the references for? 

Here a formatting error occurred. The footnote is now at the end of page 2 in order to provide additional information related to historic bark use and its documentation (lines 49-51).

Reviewer #2: The work is well written and relevant. Correct methodology. The results were analyzed correctly. Some recommendations: objetives at the end of the introduction and put specific objetives. 

We added some specific objectives towards the end of the introduction

Methodology: cite the age, diameter and height of the trees. Reason for choosing species. 

The species were selected according to their availability in the vicinity of our research institution since one of the goals was to work with local trees. Since the aim of the present work was to explore the potential of local tree barks as a raw material for panels, age, diameter and height were not recorded as well as location of the bark within the tree. However, we observed a large variation in bark thickness, curvature etc both within and between trees. This is certainly a topic of future research and has now been included in the text.

Results and discussions: to discuss the relationship of the influence of tree diameter on debarking. Mention the thickness of the barks. Amount of phloem and periderm. Show the cells of the barks. 

Thank you for your comment and your mentioned aspects are certainly interesting questions and should be addressed in the future. As written in the previous answer, the main goal of this work was a general exploration about the feasibility of using bark as a raw material to create panels. The µCT scans provide information about the cells in the barks, a detailed analysis phloem and periderm amount would require an elaborate sampling considering tree heights, ages, etc. and was beyond the scope of the present work.

---

## [Decision Letter · Decision Letter 1]

8 Jan 2023

Adhesives free bark panels: an alternative application for a waste material

PONE-D-22-24770R1

Dear Dr. Eder,

We’re pleased to inform you that your manuscript has been judged scientifically suitable for publication and will be formally accepted for publication once it meets all outstanding technical requirements.

Kind regards,

Yasir Nawab, PhD

Academic Editor

PLOS ONE

Additional Editor Comments (optional):

Reviewers' comments:

Reviewer's Responses to Questions

**Comments to the Author**

1. If the authors have adequately addressed your comments raised in a previous round of review and you feel that this manuscript is now acceptable for publication, you may indicate that here to bypass the “Comments to the Author” section, enter your conflict of interest statement in the “Confidential to Editor” section, and submit your "Accept" recommendation.

Reviewer #1: All comments have been addressed

2. Is the manuscript technically sound, and do the data support the conclusions?

Reviewer #1: Yes

3. Has the statistical analysis been performed appropriately and rigorously? 

Reviewer #1: Yes

4. Have the authors made all data underlying the findings in their manuscript fully available?

Reviewer #1: Yes

5. Is the manuscript presented in an intelligible fashion and written in standard English?

Reviewer #1: Yes

6. Review Comments to the Author

Reviewer #1: The authors have addressed the reviewers' comments in a satisfactory fashion. The manuscript can now be accepted for publication

7. PLOS authors have the option to publish the peer review history of their article (what does this mean?). If published, this will include your full peer review and any attached files.

Reviewer #1: No

---

## [Editor Report · Acceptance letter]

13 Jan 2023

PONE-D-22-24770R1 

Adhesives free bark panels: an alternative application for a waste material 

Dear Dr. Eder:

I'm pleased to inform you that your manuscript has been deemed suitable for publication in PLOS ONE. Congratulations! Your manuscript is now with our production department. 

Kind regards, 

on behalf of

Dr. Yasir Nawab 

Academic Editor

PLOS ONE